# Microemulsion Synthesis of Superparamagnetic Nanoparticles for Bioapplications

**DOI:** 10.3390/ijms22010427

**Published:** 2021-01-04

**Authors:** María Salvador, Gemma Gutiérrez, Sara Noriega, Amanda Moyano, María Carmen Blanco-López, María Matos

**Affiliations:** 1Department of Physics & IUTA, University of Oviedo, Campus de Viesques, 33204 Gijón, Spain; salvadormaria@uniovi.es; 2Istituto di Struttura della Materia-Consiglio Nazionale delle Ricerche (CNR), Monterotondo Scalo, 00016 Rome, Italy; 3Department of Chemical and Environmental Engineering, University of Oviedo, Julián Clavería 8, 33006 Oviedo, Spain; gutierrezgemma@uniovi.es (G.G.); snoriega@gmail.com (S.N.); 4Instituto Universitario de Biotecnología de Asturias, University of Oviedo, 33006 Oviedo, Spain; moyanoamanda@uniovi.es (A.M.); cblanco@uniovi.es (M.C.B.-L.); 5Department of Physical and Analytical Chemistry, University of Oviedo, Julián Clavería 8, 33006 Oviedo, Spain

**Keywords:** superparamagnetic iron oxide nanoparticles, microemulsion, co-precipitation, liposomes, lateral flow immunoassay, point-of-care analytical platforms

## Abstract

Superparamagnetic nanoparticles have seen increased potential in medical and environmental applications. Their preparation is traditionally made by the coprecipitation method, with limited control over the particle size distribution. Microemulsion methods could be advantageous due to the efficient control of the size, shape, and composition of the nanoparticles obtained. Water-in-oil (W/O) microemulsions consist of aqueous microdomains dispersed in a continuous oil phase, stabilized by surfactant molecules. These work as nanoreactors where the synthesis of the desired nanoparticles takes place through a co-precipitation chemical reaction. In this work, superparamagnetic magnetite nanoparticles with average diameters between 5.4 and 7.2 nm and large monodispersity have been synthesized through precipitation in a W/O microemulsion, with Cetyl Trimethyl Ammonium Bromide (CTAB) as a main surfactant, 1-butanol as a cosurfactant, and with 1-hexanol as the continuous oily phase. The optimization of the corresponding washing protocol has also been established since a strict control is required when using these materials for bioapplications. Their applicability in those has been proved by their encapsulation in liposomes, being tested as signal enhancers for lateral flow immunoassays by using the affinity neutravidin-biotin model system. Due to their magnetic behaviour, they were also tested for magnetic separation. These novel materials have been found to be useful for analytical applications requiring high sensitivity and the removal of interferences.

## 1. Introduction

Magnetic nanoparticles (MNPs) have lately gained ground as workhorses in potential medical, energy, and environmental applications thanks to their properties [1,2,3,4,5]. Besides their magnetism, MNPs show biocompatibility, chemical stability and dispersibility, and the possibility to be further functionalized thanks to the smart addition of different coatings in their surfaces. Since their approval to be used in humans by the Food and Drug Administration (FDA) in the United States and the European Commission in Europe, iron oxide nanoparticles have played a very important role in in-vivo diagnosis, separation, concentration, and/or manipulation of different components in some samples, i.e., blood or body fluids [6,7], contrast agents in magnetic resonance imaging (MRI) [8,9], or tracers in magnetic particle imaging (MPI) [10,11], as well as heating agents in magnetic field assisted hyperthermia [12,13,14,15,16] or drug carriers in magnetic drug targeting and magnetic gene therapies [17,18].

In biosensing, MNPs can be used as labels to detect and quantify biomolecules of interest. To do so, there are some important requirements to the successful development of a diagnostics system. First of all, particles with uniform size and magnetic properties are needed. Secondly, an efficient approach to bio-functionalize the latter to selectively target the analyte is required. Finally, a sensitive detection method is also needed. Very often, controlled agglomerates of MNPs are required in order to improve sensitivity.

Lateral flow immunoassays (LFIAs) are among the most-established point-of-care (POC) analytical platforms. The conventional immunochromatographic tests are the basis of the actual serological and antigen tests used to monitor and diagnosis of COVID-19. In recent years, research in this field is giving rise to a new generation of LFIAs relying on MNPs, and it could be considered that magnetic LFIA are emerging as powerful tool for diagnostics [19]. In this direction, previous studies from this research group showed the successful development of these kind of tests for prostatic cancer antigen, exosomes and histamine [20,21,22]. MNPs could improve the actual limits of detection because their magnetic behaviour could be used for immuno-separations, enabling the pre-concentration of analyte of interest and the removal of interferences in the sample.

If MNPs are going to be used as immunoassay labels, homogeneous nanoparticles with the desirable characteristics are required. This implies that strict control during the synthesis of MNPs should be kept. For the MNPs signal amplification, an alternative could be their controlled encapsulation using different types of nanocolloidal systems, such as nanovesicles (e.g., liposomes, niosomes or transfersomes). Nanovesicles are colloidal particles in which a concentric bilayer made up of amphiphilic molecules surrounds an aqueous compartment. Liposomes are formed by the self-assembly of phospholipids. Due to the presence of both lipid and aqueous compartments in their structure, they can be used for the encapsulation, delivery, and controlled release of hydrophilic, lipophilic, and amphiphilic compounds [23,24].

There are several methodologies to obtain superparamagnetic iron oxide nanoparticles (SPIONs). Great efforts have been made in the development of synthesis methods that allow to get a good control of and easily tune the size, shape, composition, and colloidal stability. However, this is a complicated issue since the behavior of the MNPs lies at the intersection of overlapping chemical, physical, and biological criteria. The size of the particles mostly determines their correct properties to the intended application. For example, the size can determine whether the SPIONs are used as T_1_ or T_2_ contrast agents in MRI [25], the amount of heat released in hyperthermia [26,27] or how the cells respond to their presence in in vivo intravenous applications [28,29]. Traditionally, thermal decomposition and co-precipitation have been the most used methodologies to synthesize the SPIONs. The former allows a fine control in the properties while the latter achieves already water-dispersible particles and has a higher yield [30]. Still, the particles obtained by the co-precipitation route usually lack narrow size distributions, which has a high impact on their final properties. A more recent technique with a growing interest is the water-in-oil (W/O) microemulsion method due to the efficient control of the size, shape, monodispersity and composition of the particles. Microemulsions are thermodynamic stable systems that consist of nanosized water droplets dispersed in a continuous oil medium stabilized both by surfactant and cosurfactant molecules. Indeed, this is the main advantage of this methodology as the water droplets, in which the metal precursors are dissolved, act as a nanoreactors leading to a more uniform and controlled size. Taking into account that the final properties of the SPIONs are strongly influenced by the synthesis route and its conditions due to its unique size-dependent behavior, it could be expected that the synthesis of tuned samples with narrow size distributions could result in a best control of some magnetic properties, such as, the blocking temperature and saturation magnetization.

In this work, a full methodology to synthesize controlled size SPIONs by the W/O microemulsion method (ME) is presented regarding the water to surfactant ratio, the addition of the precipitation agent, as well as the optimization of the corresponding washing protocol since the synthesis and purification conditions significantly influence the resulting structural and physicochemical properties of the SPIONs. Moreover, the ME methodology was compared to a conventional co-precipitation method performed at the same conditions. Finally, the suitability of the SPIONs obtained to be used as labels in an inductive magnetic sensor was also tested. For this purpose, all synthesized SPIONs were characterized in terms of size and shape by dynamic light scattering (DLS) (Nanozetasizer from Malvern) and transmission electron microscopy (TEM), structure/composition by x-ray powder diffraction (XRD). Once the synthesis protocol using ME was optimized, MNPs have been encapsulated in liposomes obtained by using a modified ethanol injection method (EIM). The nanovesicle size was also optimized, so they could be used as multilabel system at LFIAs for the signal enhancement. To study their feasibility as label, the neutravidin-biotin model system has been used. To do this, the magnetic liposomes have been conjugated to neutravidin to evaluate their affinity with a biotin test line immobilized at the nitrocellulose membrane. Finally, the magnetic character of the liposomes allows the magnetic separation of the biomolecules of interest from a complex matrix. The easy of the methodology could be very helpful in some analytical techniques to improve the detection of the analyte of interest.

## 2. Results and Discussion

### 2.1. Synthesis of the SPIONs by Co-Precipitation

SPIONs were synthesized by the traditional co-precipitation method, which was first demonstrated by Massart in 1981 [31]. The reaction can be simplified as:(1)$$Fe^{2+}+2Fe^{3+}+8OH^{-}\to 2Fe{\left(OH\right)}_{3}\to Fe_{3}O_{4}+H_{2}O$$

The particles obtained will be from now on denoted as CoP5.

### 2.2. Preparation and Study of the Microemulsion System

The determination of the stable W/O microemulsion was established applying the titration method at room temperature using the proper concentration of the iron salts in the aqueous solution. The ternary phase diagram for the system was constructed and is shown in Figure 1a. In this diagram, the area located between the two dotted lines is where the oil-in-water (O/W) microemulsion stable region was found.

Visual examination and conductivity measurements give information of the system that is being formulated during the titration process. As an example, Figure 1b shows the conductivity variation during the titration process as a function of the water phase content for the point marked as number 6 in Figure 1. At the beginning, the conductivity value was small, but it increased upon the addition of the water phase into the 1-hexanol-surfactant mixture since the W/O microemulsion was formed. Then, the maximum conductivity was reached for a 30% water content, which visually agreed on the change from translucent to opaque. In this low conductivity region, water was presented as dispersed droplets, and the number of the droplets increased when increasing the amount of water. The conductivity maximum in this region (30%) can be attributed to the saturation of droplets and the percolation of charges through the droplet clusters with minimum resistance. When more water phase was added to the system, it led to a phase separation and then, the conductivity lowered. After this point, the conductivity measurements begin to rise rapidly again. Transient water channels are formed when the surfactant interface breaks down during collisions or through the merging of droplets. In this case, the conductivity is mainly due to the motion of counter ions along the water channels [32]. At this point, it can be noted that the conductivity of the microemulsion system is highly influenced by its structure, such as water droplet concentration, and inverse phase phenomena [33]. The occurrence of percolation conductance reveals the increase in droplets concentration. The percolation threshold corresponds to the formation of the first infinite cluster of droplets [34].

Within the stability region obtained for the W/O microemulsions, six different formulations were selected to further continue and proceed with the synthesis of the nanoparticles. These formulations are shown in Figure 1a and their related compositions are gathered in Table 1. All the syntheses were done by adding the ammonia solution (30% wt) dropwise to the microemulsion prepared with a water phase solution containing a concentration of 0.7 M and 1.4 M of Fe (II) and Fe (III), respectively.

Microemulsion were observed using a TEM microscope. Formulations 3 and 5 are shown in Figure 2 and droplets of around 0.3 and 0.5 µm were observed, respectively.

### 2.3. Synthesis and Structural Characterization of the SPIONs

The SPIONs obtained with the six selected formulations were black once the precipitated was formed. In all cases, this coloration appeared at a pH between 9 and 10, moment in which the ammonia addition was stopped. The SPIONs obtained using the formulation 2 presented grains of a very large size whereas in the rest of the samples their appearances were close to dust. During the magnetic agitation, the samples 1, 2, 4, and 6 turned into a reddish solution, probably due to the oxidation of the freshly magnetite (Fe_3_O_4_) that could be consequently transformed into maghemite (Fe_2_O_3_). The magnetic character of the particles was checked with a magnet, and it was only present in the samples obtained with the composition of the formulation 3 and 5. Still, the non-magnetic character of these other samples suggest that another compounds such as δ-FeOOH, which are magnetic, could have been transformed into a non-magnetic α-FeOOH over time thanks to the sample being left in ambient air [35,36,37].

From now on, only the formulations 3 and 5 were selected for the subsequent experiments based on the higher stable magnetic character of the samples. The effect of varying the salt concentration in the aqueous phase and the influence of how the precipitating agent is added to the system were studied to see their effects on the resulting size and properties of the SPIONs formed (Table 2).

The proportion of the different components forming the microemulsion has a strong influence on the final nanoparticles. This has been proved before, when with some of the microemulsion compositions no magnetic nanoparticles were formed. For those were magnetic ones were obtained and to assess the influence of the water phase to surfactant ratio ($W_{0}$), samples 3 and 5 were compared. When the value of this ratio decreases, as for sample 3, the particles obtained were smaller than when this ratio takes higher values, as for sample 5. This also agrees with the particle size observed in the microemulsions (Figure 2), being 0.3 and 0.5 µm for samples 3 and 5, respectively. The formation of a thermodynamically stable microemulsion is possible when the ratio of water phase to surfactants is increased, point where the water would be rather rigid. However, if this value goes larger, more flexibility is achieved and the rate of growth increases [35,36,37]. Increasing the water content in the microeemulsion had already been suggested as a way to increase the size of the nanoparticles formed [38,39].

Regarding the concentration of iron salts in the aqueous phase, the effect seems to be the opposite: as the size of sample 5B, synthesized with the double molar concentration of precursors (but the same molar ratio between them), produced larger nanoparticles. This is in contrast with most of the literature, where an increase in the concentration of the reactants in the water solution yielded smaller particles [35,40].

Finally, to see the influence of the addition of the precipitation agent either by solution or by anther microemulsion, samples 5B and 5M were compared. The surfactant molecules in the microemulsions are dynamic, meaning that they are in continuous movement and colliding with each other. It is in these collisions where the interchange of material takes places. This dynamic process plays also an important role to control and slow down the velocity of the chemical reactions which, on the contrary, take place very fast in aqueous solutions. For sample 5B, the precipitation agent was added dropwise to the microemulsion, which means it must be diffused from the continuous phase to the water droplets. For sample 5M, the ammonia was added as another microemulsion, in this case, the droplets of both microemulsions need to collide to transfer the precipitating agent. The droplets of both microemulsions are properly mixed to promote contact trough intermicellar exchange to promote the chemical reaction (what involves fusion-fission events between the nanodroplets).

After the chemical reaction takes place inside the nanodroplets a critical number of nuclei is formed leading to further growth of the controlled size nanoparticles. Thus, nanoparticles formation process involves diffusion, collision, exchange, reaction, nucleation and growth of nuclei [41]

It has been reported by other authors that high intermicellar exchange rates lead to smaller sizes [42]. However, some differences exist on the literature related to process parameters on the final nanoparticles sizes [41]. Some authors stated that larger particles than the droplet sizes are expected by the direct ammonia addition while smaller sizes that the original droplet size can be obtained by the two separate microemulsions method [43].

The similarity in particle size obtained by both methods for this particular formulation (5B and 5M) in the present work is an indication that the ammonia diffused easily to the water droplets, containing the iron precursors, independently of the way of addition. It is important to point out that the dropwise addition took place under vigorous stirring. Therefore, it can be assumed that these initial steps, i.e., diffusion, collision, and intermicellar exchange, did not govern the overall process, leading to a similar final particle size.

TEM images in Figure 3 for these samples show the formation of spherical nanoparticles despite of a few irregular shapes and some degree of agglomeration. The images allowed to obtain the mean diameter of the particles and their size distribution. The calculated $d_{\mathrm{TEM}}$ are gathered in Table 3. Comparing the values obtained by ME method with the ones obtained by co-precipitation with the same iron salt ratio, smaller nanoparticles were obtained by ME since values registered by co-precipitation were around 9 nm. It has been previously reported that when using co-precipitation techniques, it is difficult to obtain good quality SPIONs with sizes smaller than 10 nm [44]. Moreover, the ME method allows a better control of the dispersity of the samples, as the standard deviation is almost half the value of the co-precipitation route, which has a σ value around 0.25. A narrower distribution allows a better understanding and correlation of the properties of the particles, without the influence of size distribution effects.

### 2.4. Optimization of the Washing Procedure

Although TEM images for formulations 3 and 5 showed mainly spherical shape, some other structures, such as acicular ones, were also observed (see Figure 3b,e). The latter can be attributed to a minimum amount of goethite α-FeOOH formed, but mainly, the other could be related to the formation of salts within the media components that remain in the samples. These impurities have also been found by means of the identification of the peaks in XRD, mainly being salammoniac. To avoid their presence in the samples, which could influence their properties and make the SPIONs not suitable for their final application, different washing procedures were applied and compared. By observing TEM images of the particles before and after the washing procedure, the most efficient protocol was the one that used ethanol and water in a ratio 90:10 (*v*/*v* %). The samples hereinafter are denoted as 3W and 5W and are shown in Figure 3c,f, respectively. More clear and sharp samples were observed probably due to the elimination of some remaining oil phase thanks to the washing procedure, and due to the absence of the salt impurities before mentioned.

The XRD patterns depicted in Figure 3a for samples 3W and 5W show well-defined peaks, which match the standard magnetite and clearly indicates the formation of crystalline samples with magnetite and/or maghemite composition. No other compounds were found confirming the presence of only the spinel ferrites and the effectiveness of the washing procedure. However, comparing both spectra, the sample 3W shows more sharp peaks whereas the sample 5W shows broader peak. A mixture of amorphous and nanocrystalline phases in the latter, hence, a decrease in the crystallinity could be the explanation. The average apparent crystalline size obtained by the direct application of the Thompson–Cox–Hastings pseudo-Voigt profile function, being 5.7 and 7.2 nm for the samples 3W and 5W, respectively, showed good agreement with the $d_{\mathrm{TEM}}$.

### 2.5. Magnetic Characterization of the SPIONs

The field dependence of the magnetization at room temperature (298.15 K) is shown in Figure 4b for the samples 3 and 5 before and after the washing procedure. All of them show the typical curve of superparamagnetic nanoparticles, with no hysteresis, i.e., open cycle, as can be seen in the inset of the same figure. The saturation magnetization ($M_{S}$) of the samples has been calculated by fitting the experimental data to the law of approach to saturation [45] and their values are gathered in Table 3. For samples 3 and 5, these values are 60.3 and 69.3 A·m^2^/kg, respectively, which are lower when compared to the bulk magnetite ($M_{S}$ = 92–98 A·m^2^/kg) [37].

On the other hand, there is a light decrease in the $M_{S}$ value after the washing procedure for both samples (12 and 14% for 3W and 5W, respectively, see Table 3). The possible oxidation from magnetite to maghemite could contribute negatively to the net magnetic moment of the samples and it has been observed in other procedures involving the exchange of the ligands [46,47,48].

### 2.6. Affinity Lateral Flow Assay

The hydrodynamic diameters of liposomes before and after conjugation were 182 nm (polydispersity index 0.230) and 310 nm (polydispersity index 0.233), respectively. Therefore, the biocojugation process was confirmed by the increase of the hydrodynamic diameter since the neutravidin attachment on its surface causes this increase.

Once liposomes have been functionalized, to perform the dipstick affinity assay, 10 µL of the magnetic liposomes functionalized with neutravidin and 90 µL of running buffer were added into the microtube. The strip was immersed in the liquid sample, which flows by capillarity. As it is well known, the neutravidin protein recognizes specifically the biotin molecules. Therefore, only the magnetic liposomes functionalized with neutravidin would be retained at test line (sample in Figure 5). Figure 5 shows a blank affinity assay with bare magnetic liposomes and a sample strip with magnetic liposomes bioconjugated to neutravidin. With the blank assay, the magnetic liposomes were not attached to the biotin test line. On the contrary, when the magnetic liposomes were conjugated to neutravidin, a dark brown test line can be seen, confirming the affinity bond between biotin and the bioconjugated neutravidin. Biotin is a low molecular weight molecule, and in order to be immobilized at the test line, it is better to use a biotin-BSA (bovine serum albumin) complex. Figure 5 also shows that there are non-specific interactions between bare magnetic liposomes and biotin-BSA test line. This is very important for further applications of the magnetic liposomes.

### 2.7. Immuno-Separation

In addition, the magnetic liposomes have an additional advantage, since they can be used for immuno-separation. For this approach, magnetic liposomes have to be functionalized with a ligand that specifically recognizes the analyte of interest. Once the labels are attached to the analyte, the resulting conjugate can be separated easily with a conventional magnet in less than three minutes (Figure 6c). After the magnetic separation, the supernatant remains very clear (almost transparent). This result confirms the efficiency, rapidity, and simplicity of the magnetic liposomes separation with a conventional magnet. Then, the supernatant is discarded to remove other interferences, and finally the pellet is resuspended in the buffer required. Figure 6 summarizes the several steps of immune-separation schematically and with real pictures obtained with the magnetic liposomes synthesized.

The immune-separation process in combination with LFIAs would have great impact for the detection of analytes in complex matrix, without the need for previous purification steps. With this combination, it is possible to reduce the time of sample preparation with the added advantage of preconcentration to lower the limits of detection.

## 3. Materials and Methods

### 3.1. Materials

Ferric Chloride Hexahydrate (FeCl_3_∙6H_2_O) and ammonia 30% (NH_3_) were supplied by Panreac AppliChem (Barcelona, Spain). Ferrous Chloride Tetrahydrate (FeCl_2_∙4H_2_O) was supplied by J.T. Baker (Phillipsburg, AL, USA). Cetyl Trimethyl Ammonium Bromide 99% (CTAB), 1-butanol (min. 99%), hydrochloric acid 38% (HCl), ethanol (95%), phosphotungstic acid hydrate (99.995%), 1-ethyl-3-[3-di- methylpropyl]carbodiimide (EDC), *N*-hydroxysuccinimide (NHS), bovine serum albumin (BSA), biotin-conjugated bovine serum albumin (BBSA) and Tween20 were supplied by Sigma-Aldrich (Darmstadt, Germany). 1-Hexanol was supplied by Merk (Darmstadt, Germany). Nitric Acid, min. 69.5% (HNO_3_) was supplied by Scharlab, S.L. (Barcelona, Spain). Sodium Hydroxide (NaOH) was supplied by EMSURE (Darmstadt, Germany). Neutravidin protein was supplied by Thermo Fischer Scientific (Waltham, MA, USA). None of the reagents was further modified before its use.

For synthesis of magnetic liposomes, Phosphatidylcholine (PC) (predominant species: C_42_H_80_NO_8_P, MW = 775.04 g/mol) from soybean (Phospholipon 90G) was a kind gift from Lipoid (Ludwigshafen am Rhein, Germany). Cholesteryl hemisuccinate (Cho) (C_31_H_50_O_4_, MW = 486.73 g/mol) was purchased from Sigma Aldrich (USA). All membrane components were dissolved in absolute ethanol (Sigma Aldrich, Darmstadt, Germany).

For affinity biotin–neutravidin lateral flow assay, glass fiber membrane (GFCP001000) used as sample pad and backing cards (HF000MC100) were supplied by Millipore (Darmstadt, Germany). The other necessary materials to manufacture the strips were nitrocellulose membranes (UniSart CN95, Sartorius, Spain) and absorbent pads (Whatman, Madrid, Spain).

An IsoFlow reagent dispensing system (Imagene Technology, Lebanon, NH, USA) was used to immobilize the test line. A guillotine Fellowes Gamma (Madrid, Spain) was used to cut the individual strips.

The sample buffer used to run the strips consisted of 10 mM phosphate buffer (PB) pH 7.4 with 0.5% Tween-20 and 1% BSA.

### 3.2. Synthesis of the SPIONs by Co-Precipitation

SPIONs were synthesized by co-precipitation, which is the most widely applied synthesis technique [30]. Among all the parameter that can influence the properties of the particles, the ratio of iron salts has been shown to vary their sizes [49]. With this purpose, a 100 mL solution with a molar ratio of iron salts Fe^II^/Fe^III^ of 0.5 were prepared, containing 0.01 M of HCl to avoid further oxidation of the Fe^II^. Then, 30 mL of ammonia solution (30% wt) was added and a black precipitated instantly appeared. The solution was left to rest for 2 h and then washed with distilled water several times assisted by a permanent magnet.

### 3.3. Preparation of the W/O Microemulsions

W/O microemulsions were formulated using CTAB acts as the cationic surfactant and 1-butanol as co-surfactant using a constant weight ratio 3:2 (CTAB:1-butanol). The water phase consisted of a solution that contained the iron precursors in a 2:1 molar ratio (Fe^II^/Fe^III^ = 0.5). It was prepared by dissolving an appropriate amount of the chloride salts aforementioned being homogenized by magnetic stirring. A 0.01 M HCl solution was added to avoid Fe(II) oxidation.

1-hexanol was used as the oily phase. The microemulsion stability region was determined by the titration method [50] together with conductivity measurements at room temperature. The different steps involved in the followed procedure are shown in Figure 7. The first step (1) was the addition of proper amounts of CTAB, 1-Butanol and 1-Hexanol, corresponding to each one of the formulations tested, obtaining a solution with a white and opaque appearance. When adding the water phase in a low percentage, the solution was still opaque but with an orange coloration (2). By increasing the water phase percentage, the W/O microemulsion stability region was reached when samples turned into a translucent appearance. (3) The end of microemulsion stability region was detected when the solution became opaque again with further water phase addition, indicating the presence of the multiphase region (4, 5, 6, 7). The conductivity changed sharply when the W/O emulsion turns into an oil-in-water O/W emulsion (8) [51,52].

### 3.4. Synthesis of the SPIONs by the Microemulsion Method

The synthesis of the SPIONs was also performed through the co-precipitation of the iron salts present in the microemulsion water droplets by the addition of a precipitating agent while stirring. From the W/O microemulsion stability region determined in the ternary diagram, six different formulations were selected. These microemulsions were prepared and left to rest for a while until their appearance was totally translucid. To produce the nanoparticle precipitation, ammonia (30% wt) was added as the precipitating agent. For this purpose, two different methods were used: (i) the direct dropwise addition of an ammonia solution and (ii) the addition of another W/O microemulsion prepared with the same formulation (i.e., same amount of CTAB, 1-butanol and 1-hexanol and same amount of water phase) except for the water phase that consisted now of the ammonia solution (30% wt) (Figure 8). In both cases, the ammonia (either in solution or in the water droplets of the W/O microemulsion) was added dropwise into the microemulsion system upon vigorous stirring with the SilentCrusher M Homogenizer (Heidolph 8F) set at 6500 rpm. When a black precipitate appeared, the solution was left for two hours under magnetic stirring.

### 3.5. Optimization of the Synthesis and Washing Procedure

Bioapplications of the SPIONs require a strict control of the material being used as well as its properties. At this point, the presence of some impurities or compounds in the reaction media could influence on the SPIONs behavior. In addition, the procedure used to get rid of these impurities, should not change either the nature of the particles nor their designed properties. A washing procedure has been optimized by using different ethanol:water volumetric ratios being 100:0, 90:10, 75:25, 50:50, 25:75, and 0:100. All the samples were washed five times with each one of these solutions assisted by magnetic separation. The samples were characterized before and after the washing procedure in order to confirm if this step has any influence on their resulting magnetic properties.

### 3.6. Physicochemical Characterization of the SPIONs

The synthesized SPIONs were characterized by TEM. A small amount of an aqueous diluted suspension of the samples was placed into a copper-grid-supported transparent carbon foil and examined (MET JEOL-2000 EX-II). The mean particle size and distribution of the particles were evaluated by measuring the dimension of at least 100 particles in the images obtained. Afterward, the data was fitted to a log-normal distribution to obtain the mean size ($d_{\mathrm{TEM}})$ and the standard deviation (σ) [53]. Microemulsions were also analyzed by TEM. Samples were negatively stained with a 1% phosphotungstic acid solution.

The concentration of the samples was either obtained by thermo-gravimetrical analysis (TGA) or by the iron content measured with an inductively coupled plasma mass spectrometer (ICP-MS) HP 7500ce Agilent Technologies. For this purpose, the samples were digested with aqua regia to oxidize the possible organic remaining and to dissolve the particles.

The lyophilisation of the SPIONs was carried out in order to obtain a powder of the samples. The crystal structure was identified by XRD performed in Philips X’ Pert Pro Panalytical diffractometer using CuKα1,2 radiation in a Bragg–Brentano reflection configuration. In order to obtain the instrumental broadening contribution and deconvolute the line profile function, pure Iron (II;III) oxide (Puratronic^®^) with high crystallinity, was used as external reference. The estimation of the crytalline domain size of the magnetite-nano grains was obtained using the FullProf program [54] and the direct application of the Thompson-Cox-Hastings pseudo-Voigt profile function expressed by a weighted sum of Gaussian and Lorentzian [55].

Size and particle size distribution and polydispersity index (PDI) were analyzed by DLS with Zetasizer Nano S instrument (Malvern Instruments).

### 3.7. Magnetic Characterization of the SPIONs

The magnetization curves at room temperature set up at 298.15 K of the samples were obtained using a vibrating sample magnetometer (VSM) EV9 equipped with an electromagnet producing fields up to ±2.2 T. The measurements were done in powder form, in field steps of 0.05 T and the results were normalized to the magnetic phase.

### 3.8. Synthesis of Magnetic Liposomes

For magnetic liposomes preparation, PC was dissolved in 20 mL of absolute ethanol in a concentration of 8 mg/mL. Cholesterol hemisuccinate was also added into the organic phase (0.1% of the total mass of PC). Then, the organic solution was injected, with a syringe pump (KD Scientific, Holliston, MA, USA) at a flow of 120 mL/h, into 50 mL of Milli-Q water that was kept at 60 °C and stirred at 500 rpm. Once the vesicles were formed, ethanol was removed at 40 °C under reduced pressure (90 kPa) in a rotary evaporator.

In order to perform the encapsulation, 200 µL of the solution containing the SPIONs was added in the aqueous phase before the injection was made.

The resulting samples were further sonicated for 15 min (CY-500 sonicator, Optic Ivymen System, Spain), using 55% amplitude, 500 W power and 20 kHz frequency.

Finally, nanovesicles containing the SPIONs were purified by gel permeation chromatography in order to eliminate the non-encapsulated nanoparticles. For this purpose, a Sephadex G-25 Superfine column (HiTrapTM desalting columns, GE Healthcare Life Sciences, UK) was used/ using a gravity elution PD Column (V0 = 2.5 mL) packed with Sepharose CL-4B (both from, GE Healthcare Life Sciences).

### 3.9. Preparation of Lateral Flow Strips

The strips were based on a dipstick format which comprise four paper-based components: absorbent pad, nitrocellulose membrane, sample pad, and plastic backing card. To prepare the strips, firstly, the nitrocellulose membrane was assembled into a plastic backing card. Then, the test line was dispensed using a solution that contain 1 mg/mL of biotin-BSA. For this, an IsoFlow dispenser was used with a rate of 0.100 µL/mm. After dispensing, membrane was kept at 37 °C at least 30 min to ensure the immobilization of test line. Once the test line was dried, the other components of the strips (absorbent pad and sample pad) were overlapped to nitrocellulose membrane around 2 mm using the adhesive part of the backing card. Finally, the strips were cut individually with a width of 5 mm and preserved at room temperature with desiccant bags until its use.

### 3.10. Functionalization of Magnetic Liposomes

To study the feasibility of magnetic liposomes to be used as labels in LFIA, they were bio-conjugated to a model protein called neutravidin. The magnetic liposomes contain carboxyl groups on its surface since a portion of cholesterol used during its synthesis was modified with this functional group. Thus, they can be bond to amine groups of neutravidin by carbodiimide coupling chemistry. To establish this covalent bond, 1.5 mg of EDC and 3 mg of NHS were added to 1.5 mL of suspension containing the magnetic liposomes to activate the carboxyl groups shaking for 40 min. After activation step, the solution was centrifuged at 6800× *g* for 10 min. Then, 1.5 mL of supernatant was discarded, and the pellet was dissolved in 100 µL of PB. For the bio-conjugation, 0.5 mg of neutravidin was added to activated magnetic liposomes and they were shaken continuously for 2 h. After conjugation, the solution was centrifuged at 6800× *g* for 10 min. Then, 200 µL of supernatant was discarded, and the pellet was dissolved in 100 µL of PB. Finally, the residual functional groups and the liposomes surface were blocked with 100 µL of a solution containing 1 mg/mL of glycine. After blocking, the solution was centrifuged at 6800× *g* for 10 min. Then, 200 µL of supernatant was discarded, and the pellet was dissolved in 100 µL of PB.

To demonstrate that neutravidin is attached on surface of liposomes hydrodynamic diameter was measured using Dynamic Light Scattering. For these measurements, a Zetasizer Nano ZS ZEN3600 (Malvern Instruments, Malvern, UK) equipped with a solid-state He–Ne laser (λ = 633 nm) was used to measure size distribution. 30 measurements of the backscattered (173°) intensity were carried out at 25 °C and averaged. For data analysis, Zetasizer software version 7.03 was used.

## 4. Conclusions

The great number of technological applications of the SPIONs has increase the need of suitable synthesis protocols that allow to tune their characteristics for a particular purpose. Microemulsions have been proposed as an alternative method to the classical co-precipitation methods. They provide a better control of the size distribution of the particles, and hence, of their final properties. Here, a synthesis route based on a W/O microemulsion that allows to obtain superparamagnetic SPIONs have been developed. The control in the size and the narrower size distribution leads to more homogenous properties, including a higher value of the saturation magnetization, but specially, showing optimal results for further development and application of these SPIONs as labels for LFIAs. For their use as labels, SPIONs have been encapsulated in liposomes to amplify the magnetic signal in LFIAs. The surface of the magnetic liposomes has also been modified with carboxyl groups for their ulterior functionalization. Their feasibility as labels have been demonstrated using the affinity neutravidin-biotin model. Their potential for magnetic separation as previous step to detection has also been demonstrated.

## Figures and Tables

**Figure 1 ijms-22-00427-f001:**
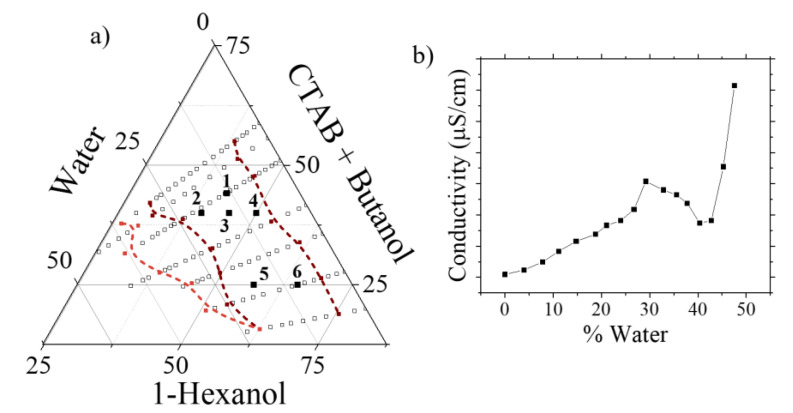
(**a**) Ternary diagram of the CTAB-butanol–hexanol–water system showing the composition of the different microemulsions used for the synthesis of the SPIONs, denoted as 1, 2, 3, 4, 5 and 6, all of them within the microemulsion stability region. (**b**) Conductivity measurements as a function of the water content for the point 6 in the CTAB-butanol-hexanol microemulsion system.

**Figure 2 ijms-22-00427-f002:**
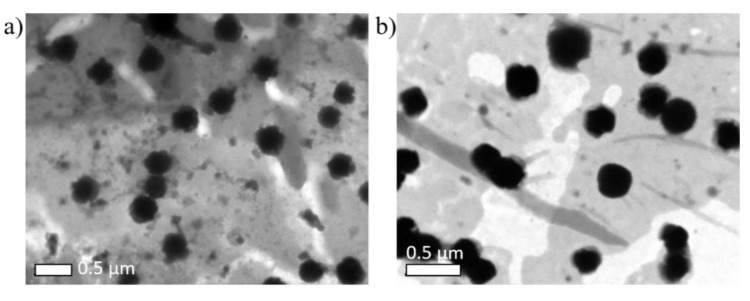
TEM images of the W/O microemulsions negatively stained with 1% phosphotungstic acid. Microemulsion with a composition corresponding to the formulations (**a**) 3 and (**b**) 5 in the ternary diagram of the CTAB-butanol–hexanol-water system.

**Figure 3 ijms-22-00427-f003:**
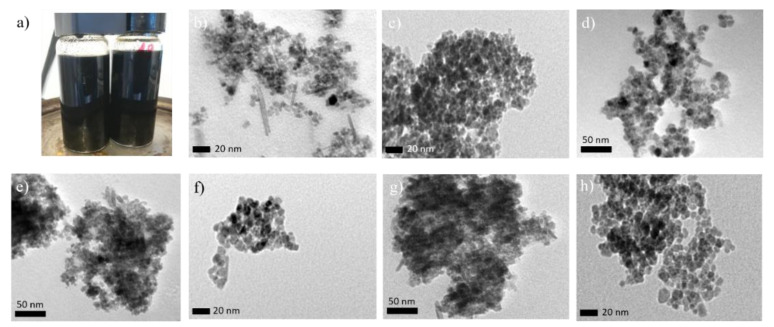
(**a**) Picture of the SPIONs synthesized and TEM images of the samples (**b**) 3, (**c**) 3W, (**d**) 5B, (**e**) 5, (**f**) 5W, (**g**) 5M and (**h**) CoP5.

**Figure 4 ijms-22-00427-f004:**
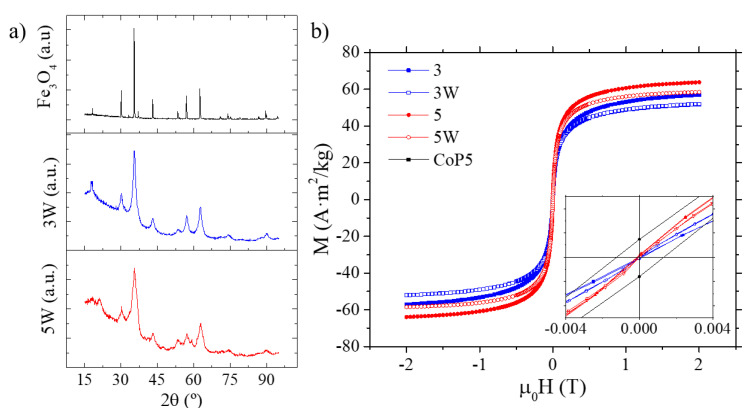
(**a**) XRD spectra obtained from the samples 3W and 5W compared with the magnetite pattern Puritronic^®^. (**b**) Magnetization curves at room temperature for the samples 3 and 5 before (3, 5) and after the washing procedure (3W, 5W). Inset: central area detail of the magnetization curves for the samples obtained by co-precipitation, CoP5, and ME.

**Figure 5 ijms-22-00427-f005:**
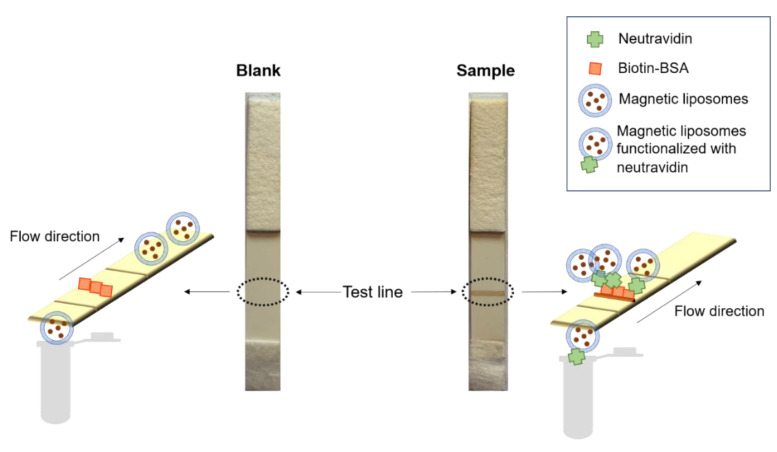
Scheme of affinity lateral flow assay using magnetic liposomes as labels.

**Figure 6 ijms-22-00427-f006:**
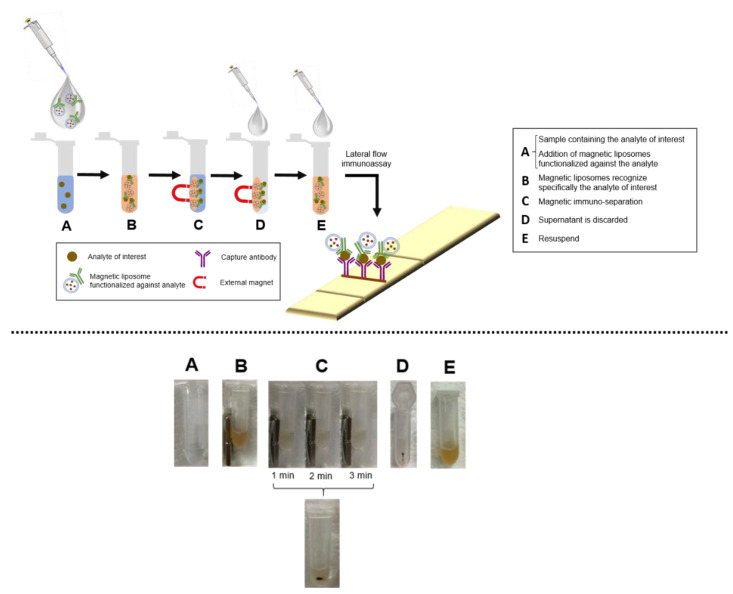
Process of immuno-separation, including the different steps schematically and with photographs. (**A**) Solution with the analyte. (**B**) Immuno -separation with a conventional magnet using magnetic liposomes. (**C**) Immuno-separation after 1 min, 2 min and 3 min. (**D**) Pellet after immune-separation. (**E**) Pellet resuspended with buffer.

**Figure 7 ijms-22-00427-f007:**
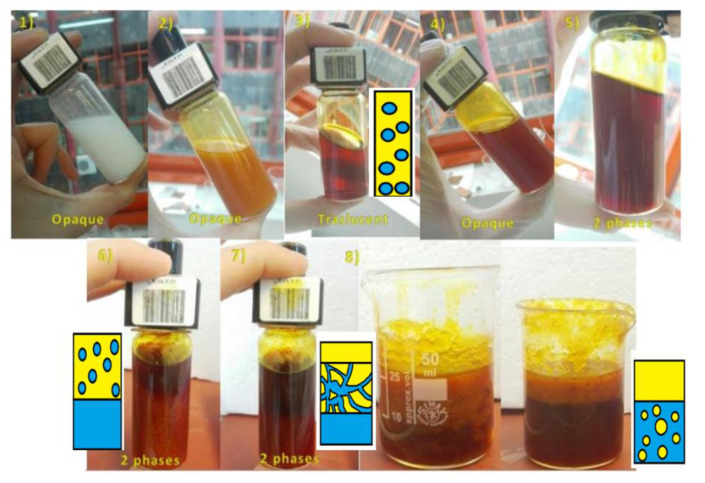
Evolution of the microemulsion system CTAB-1butanol-1hexanol with the increasing water phase content. A schematic view of the vial is showed for better understanding of the phases formed while the aqueous phase was added during titration.

**Figure 8 ijms-22-00427-f008:**
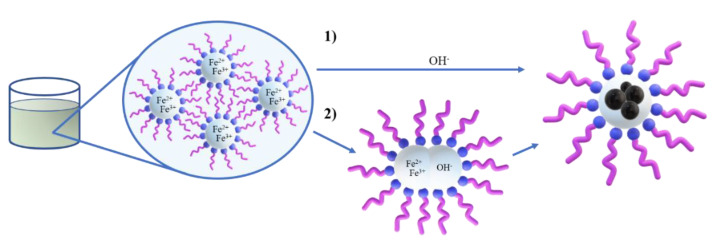
Schematic view of the synthesis of the magnetic nanoparticles studied. Water-based solutions of the iron precursors are used to form a microemulsion in the oil phase. The precipitation of the particles is achieved by means of the dropwise addition of either (**1**) the basic ammonia solution or (**2**) another microemulsion containing the ammonia.

**Table 1 ijms-22-00427-t001:** Microemulsion composition (% wt) of the six formulations selected within the microemulsion stability region.

Sample	Microemulsion Composition (%wt)			
	CTAB	1-Butanol	1-Hexanol	Water
1	26.5	17.7	42.5	13.3
2	24	16	40	20
3			45	15
4			50	10
5	15	10	57	18
6			65	10

**Table 2 ijms-22-00427-t002:** Nanoparticles samples synthesized by the ME method with their compositions (% wt). Molar composition of the aqueous phase, and the way of adding the precipitation agent.

Sample	Microemulsion Composition (% wt)				[Salt]		Precipitating Agent
	CTAB	1-Butanol	1-Hexanol	Water	M Fe^3+^	M Fe^2+^	
3	24	16	45	15	1.4	0.7	Solution
5	15	10	57	18	1.4	0.7	Solution
5B					0.7	0.35	Solution
5M					0.7	0.35	Microemulsion

**Table 3 ijms-22-00427-t003:** Mean diameter *d*_TEM_ obtained from TEM images and its standard deviation *σ*_TEM_, and saturation magnetization *M_S_* obtained from the magnetization curves at room temperature.

Sample	*d*_TEM_(*σ*_TEM_) (nm)	*M_s_* emu/g Fe_3_O_4_
3	5.4(0.19)	60.3
3W		53.7
5	6.6(0.11)	69.3
5W		60.5
5B	7.1(0.17)	65.7
5M	7.2(0.15)	49.7
CoP5	9.8(0.23)	58.8

## Data Availability

Data is contained within the article.

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
