# Peer review of "Microemulsion Synthesis of Superparamagnetic Nanoparticles for Bioapplications"

_ijms, 2021, doi:10.3390/ijms22010427_

Round 1

Reviewer 1 Report

The authors of the manuscript proposed a paper dedicated to the microemulsion method for the synthesis of magnetic nanoparticles. The manuscript is well written but the authors did not avoid some inaccuracies.

1. In Table 2 there is no “average particle size diameters (nm) obtained from TEM images dTEM together with the standard deviation of their distributions σ” presented as suggest the authors.

2. Very poor quality of TEM images – it should be changed for better quality, so the reader should verify the morphology, shape and diameter of the nanoparticles.

3. The research presented in the manuscript is not easy to follow. Some new abbreviations (which describe the nanoparticles) which are presented in the Figures or in tables are not clearly described in the text (e.g. CoP5 presented in the Figure 3 – what is that?).

4. In the immobilization method of neutravidin on the liposomal encapsulated nanoparticles there is lack of characterisation of the samples – e.g. some FT-IR should be done to show the differences between starting material and the product.

5. Spelling should be checked – the authors did not avid spelling mistakes e.g. line 379 there is “my” should be “by” etc.

Also as a rewiever I have some questions:

1. Did the researchers study the stability of microemulsion during precipitation process? Do addition of ammonia influence on stability of the microemulsions?

2. Why there is no difference between the size of nanoparticles which were precipitated by ammonia and by ammonia microemulsion? There is no discussion about it.

Reviewer 2 Report

Report of Manuscript ijms-1054254 for IJMS

Title: Microemulsion synthesis of superparamagnetic nanoparticles for bio-applications by M. Salvador et al.

In this work, the authors studied Superparamagnetic Iron Oxide Nanoparticles (5.4-7.2 nm of diameter) and monodispersity synthesized through precipitation in a water-in-oil microemulsion. The optimization of the corresponding washing protocol has been established since a strict control is required when using these materials for bio-applications. Synthesized nanoparticles were characterized though DLS TEM and XRD. These materials have been found useful for analytical applications requiring high sensitivity and removal of interferences.

The paper is well-written, the English is good and I think that this work could be of interest for the field of Nanomaterials and Superparamagnetic nanoparticles. The paper is technically sound, but it is limited and preliminary. The work is well structured and the proposed goals were achieved. The manuscript contains new and preliminary information to justify publication. The methods described comprehensively. The interpretations and conclusions justified by the results.

Major Issue

  • This reviewer is skeptical about the editorial location of this manuscript. It appears that the novelty in the present manuscript is the development of a new way to synthesize nanoparticles. Would “IJMS” be the best forum to discuss these results? The authors might consider a more suitable journal devoted to “material sciences" (i.e. "Nanomaterials - MDPI"). Please justify.
  • The manuscript is well presented, but needs improvements regarding the chemical-physical motivations. Please, make a check in the whole manuscript.
  • I think the more clear flow of an article is: introduction, materials and methods, results, conclusions. Why did the authors choose to put "M&M" at the end?
  • In all part of the main text, different minor typo corrections that should be performed.

Minor Issue

Line 27 – The authors should specify "CTAB".

Line 31 – Delete “Nanozetasizer from Malvern” in the abstract

Introduction - I think the introduction could be shortened by focusing more on the objectives of this manuscript.

Page 2 - The authors could enrich this part of the introduction with more recent references on the use of magnetic nanoparticles. I suggest adding and seeing for example: “https://doi.org/10.3390/nano10101919”; “https://doi.org/10.3390/app10207322” https://doi.org/10.3390/nano10112310 and others.

Figure 1 - This figure must be enlarged in size and fonts.

Lines 141-142 – “very sensitive”. It is not clear what the authors mean by "very". Quantify this.

Figures 2 and 3 - These figures are without metrics. Please add.

Line 170 - Authors should add a more motivational sentence on the choice of preparations "3" and "5".

Section 2.5 - Specify better what is meant by "room temperature". Have measurements been carried out at low temperatures?

Section 3.7 - This section needs to be enriched with more details.

I look forward to receiving answers to my doubts soon and to be able to read a new updated and improved version of this interesting manuscript.

Round 2

Reviewer 1 Report

The manuscript was improved properly according to reviewer's comments.

It can be accepted in the present form.

Reviewer 2 Report

In general, I am pleased with the authors responses to my concerns, and the general improvements made to the manuscript after the initial assessment.

I am happy to recommend the manuscript for publication in the current form.

Note: "acronyms", in my opinion, must always be defined the first time they are used in the text.